# HyperKAN: Kolmogorov–Arnold Networks Make Hyperspectral Image Classifiers Smarter

**DOI:** 10.3390/s24237683

**Published:** 2024-11-30

**Authors:** Nikita Firsov, Evgeny Myasnikov, Valeriy Lobanov, Roman Khabibullin, Nikolay Kazanskiy, Svetlana Khonina, Muhammad A. Butt, Artem Nikonorov

**Affiliations:** 1Samara National Research University, Samara 443086, Russia; mevg@geosamara.ru (E.M.); valery2698@mail.ru (V.L.); khabibullin.rm@ssau.ru (R.K.); kazanskiy@ssau.ru (N.K.); khonina.sn@ssau.ru (S.K.); butt.m@ssau.ru (M.A.B.); artniko@gmail.com (A.N.); 2Adyghe State University, Maykop 385000, Russia

**Keywords:** Kolmogorov–Arnold networks, hyperspectral imaging, classification, transformers, convolutional neural networks

## Abstract

In traditional neural network designs, a multilayer perceptron (MLP) is typically employed as a classification block following the feature extraction stage. However, the Kolmogorov–Arnold Network (KAN) presents a promising alternative to MLP, offering the potential to enhance prediction accuracy. In this paper, we studied KAN-based networks for pixel-wise classification of hyperspectral images. Initially, we compared baseline MLP and KAN networks with varying numbers of neurons in their hidden layers. Subsequently, we replaced the linear, convolutional, and attention layers of traditional neural networks with their KAN-based counterparts. Specifically, six cutting-edge neural networks were modified, including 1D (1DCNN), 2D (2DCNN), and 3D convolutional networks (two different 3DCNNs, NM3DCNN), as well as transformer (SSFTT). Experiments conducted using seven publicly available hyperspectral datasets demonstrated a substantial improvement in classification accuracy across all the networks. The best classification quality was achieved using a KAN-based transformer architecture.

## 1. Introduction

Hyperspectral imaging (HSI) is a three-dimensional structure with two spatial and one spectral coordinates. Unlike RGB and multispectral images, HSIs are recorded at a much higher spectral resolution. Each material, which has a unique spectral signature that serves as a “fingerprint”, can be uniquely identified in HSI [1,2]. This allows one to perform analysis tasks such as classification of objects on HSI, segmentation, clustering, object detection, tracking changes in time series, etc., with higher quality compared to conventional RGB or multispectral images. The core objective of HSI analysis is pixel-wise classification, where each pixel in an image is assigned a specific class label based on its distinct spectral characteristics. This kind of task is widely in demand in remote sensing, particularly for agricultural problems [3], environmental monitoring [4], cartography [5] (for example, determining the boundaries of reservoirs [5] and glaciers [6]), etc.

Since HSIs contain both spatial and detailed spectral information about the captured scene, analysis, particularly the classification of such images, requires special tools [7]. To solve problems of HSI analysis, classical methods were used, namely, the spectral angle mapper (SAM) [8], spectral indices (NDVI, NDWI, etc.) [9], and others. In addition, general-purpose machine learning methods were utilized, such as SVM [10], decision trees [11], random forest [12], etc. However, such methods do not take into account spatial information contained in the HSI and, moreover, cannot extract deeper dependencies in spectral data. To extract deep features from spectral data more efficiently, deep architectures have been used. These architectures were based on convolutional neural networks (CNNs), namely, 1D networks [13], as well as based on the increasingly popular transformer architectures (see, for example, [14]).

To take into account spatial information, it is necessary to use deep learning tools based on spatial or spatial–spectral feature extractors. Such extractors were built based on two-dimensional convolutional layers (see, for example, 2D networks [15]) or three-dimensional convolutional layers (see 3D networks [16]). This allows a significant increase in classification quality in some cases compared to approaches that use exclusively spectral data.

Regrettably, deep neural networks (DNNs) necessitate vast amounts of data for effective training [17]. In HSI, labelling is highly labor-intensive due to its reliance on ground-based measurements, which limits the practical application of DNNs. Consequently, the ability of a neural network to learn from limited data is a crucial factor in making DNNs viable for HSI analysis [18]. Another important aspect related to HSIs is the consequence of the nature of such images. Since HSIs are recorded with very high spectral resolution, adjacent spectral bands are highly correlated [19]. This causes a strong redundancy of HSIs. As a result, the spectral space of an HSI may turn out to be sparse. Additionally, semantic classes of objects depicted in the scene often exhibit a wide variability due to many factors, such as the influence of the atmosphere, uneven illumination, multiple light reflections, etc. Considering the aforementioned limited capabilities of manual labelling, these reasons create prerequisites for manifesting the curse of dimensionality phenomenon. This, in turn, leads to a deterioration of the classification quality by traditional techniques and determines the need to search for new HSI classification methods that can overcome these challenges [20].

Shenming et al. introduced a novel hyperspectral image classification method that integrated 2D Gabor filtering with random patch convolution (GRPC) for enhanced spatial–spectral feature extraction [20]. The proposed approach began by applying dimensionality reduction techniques, namely, principal component analysis (PCA) and linear discriminant analysis (LDA), to simplify the data and focus on key information. The reduced-dimensional images were then processed using a Gabor filter to extract edge textures and spatial features. Subsequently, the spatial information derived from the Gabor filter was convoluted with random patches to extract the spectral features. The final step involved fusing the spatial and multi-level spectral features, which were then used for classification through a support vector machine (SVM) classifier. To validate the effectiveness of this method, experiments were performed on three benchmark hyperspectral datasets: Indian Pines, Pavia University, and Kennedy Space Center. The proposed approach achieved impressive overall classification accuracies of 98.09%, 99.64%, and 96.53%, respectively—surpassing the performance of several other comparison methods [20].

The deep learning algorithms are rapidly evolving, with significant advancements being made in new architectural designs. One example is networks based on the Kolmogorov–Arnold theorem [21], which have shown high efficiency in deep learning tasks. Such networks are called Kolmogorov–Arnold networks (KANs) which leverage this theorem by structuring their layers to capture complex, high-dimensional relationships through compositions of basic functions. This architecture is particularly effective in modeling intricate patterns and dependencies in data, making it highly suitable for tasks involving non-linear and multi-dimensional data, such as HSI classification and other complex recognition problems. By integrating elements such as batch normalization and specialized activation functions, KANs enhance learning efficiency and improve performance in various machine learning applications. The innovative design of KANs enables them to outperform traditional neural network models, especially in scenarios requiring robust handling of diverse and high-dimensional datasets.

For example, Vaca-Rubio et al. demonstrated that KANs significantly outperform traditional multi-layer perceptrons (MLPs) in satellite traffic forecasting tasks, achieving more accurate predictions with substantially fewer learnable parameters [22]. Cheon presented KANs as a promising alternative for efficient image analysis in remote sensing, highlighting their effectiveness in this domain [23]. Additionally, Wang et al. showed that KANs greatly surpass MLPs in accuracy and convergence speed for solving numerous partial differential equations in computational solid mechanics, though they face challenges with complex geometry problems [24]. Bresson et al. presented preliminary results that suggest while KANs are comparable to MLPs in classification tasks, they offer a distinct advantage in graph regression tasks [25].

The purpose of this paper is to demonstrate the effectiveness of replacing traditional layers with KAN analogs to enhance HSI classification. The contributions of this paper are as follows:

Three KAN-based blocks have been introduced to replace conventional classification and feature extraction layers in fully connected, convolutional, and transformer-based HSI classification architectures. The findings indicate that these KAN blocks lead to significant performance improvements, and it has been shown that batch normalization positively impacts classification accuracy when using KAN layers. Six neural network architectures—1D-CNN [26], 2D-CNN [27], 3D-CNN [28], 3D-CNN [29], NM3DCNN [30], and SSFTT [31]—have been modified by incorporating the proposed KAN blocks. These KAN-based versions of the networks have been thoroughly investigated and made available as open-source [32]. We validate the advantages of the six KAN-based architectures over their traditional counterparts using seven publicly available hyperspectral datasets, addressing the challenge of pixel-wise HSI classification.

The paper is structured as follows: Section 2 details the classical neural network blocks and their KAN analogs, explores the transformation of six different neural network architectures using KAN blocks, and presents the proposed KAN-based architectures. Section 3 describes the seven hyperspectral datasets used in the study and outlines the experimental conditions. In Section 4, we provide a detailed presentation and in-depth analysis of the experimental results for all the neural networks examined in our study. The paper concludes with a summary of findings and a list of references. Our code is published at https://github.com/f-neumann77/HyperKAN.

## 2. Materials and Methods

Typical neural network architectures for HSI classification can be conceptualized using a two-block scheme:Deep feature extractor;Classification block (commonly based on either an MLP or a single linear layer).

While traditional MLPs utilize fixed activation functions for each node (neuron), KANs introduce a novel approach by implementing learnable activation functions on the edges (weights) of the network. Unlike MLPs, KANs do not employ linear weights. Instead, each weight parameter is represented by a one-dimensional function, which is parameterized as a spline. This innovative design allows KANs to achieve superior accuracy and interpretability compared to conventional MLPs [21]. Furthermore, integrating KANs into established architectures, such as U-Net [33], has been shown to enhance performance metrics significantly.

In this study, we propose modifications to both the classification block and the feature extraction block by incorporating KAN-based layers. The investigation encompasses several architectures, including simple MLPs, as well as one-dimensional, two-dimensional, and three-dimensional convolutional models and transformer-based architectures. We hypothesize that these enhancements will lead to improved classification performance.

### 2.1. KAN Blocks

Let us examine the modifications in detail. The classification block was restructured by replacing its original configuration with KAN layers, leading to what we will refer to as the linear-KAN block [21]. Similarly, in the feature extraction block, the traditional linear layers used in attention mechanisms, or the convolutional layers may be substituted with their KAN-based counterparts. These adjustments are designed to harness the advantages of KANs, potentially leading to enhanced model performance and interpretability.

#### 2.1.1. Linear-KAN Block

Let $W_{i}$ be a weight matrix of the i-th layer of MLP network containing L fully connected layers and σ be a non-linear activation function. The MLP model can be presented as [21]
$$MLP(x)=(W_{L-1}\circ \sigma \circ W_{L-2}\circ \sigma \circ \ldots \circ W_{1}\circ \sigma \circ W_{0})(x),$$
where symbol ∘ denotes function composition, such as (f∘g)(x)=f(g(x)).

A similar equation for the Kolmogorov–Arnold network containing L layers can be represented as [21]
(1)$$KAN(x)=(\Phi_{L-1}\circ \Phi_{L-2}\circ \ldots \circ \Phi_{0})(x).$$

Here, $\Phi_{l}$ is the matrix of $n_{l+1}\times n_{l}$ activation functions:(2)$$\Phi_{l}=\left(\begin{array}{cccc} \phi_{l,1,1}(\cdot ) & \phi_{l,1,2}(\cdot ) & \cdots & \phi_{l,1,n_{l}}(\cdot )\\ \phi_{l,2,1}(\cdot ) & \phi_{l,2,2}(\cdot ) & \cdots & \phi_{l,2,n_{l}}(\cdot )\\ \cdots & \vdots & & \vdots \\ \phi_{l,n_{l+1},1}(\cdot ) & \phi_{l,n_{l+1},2}(\cdot ) & \cdots & \phi_{l,n_{l+1},n_{l}}(\cdot ) \end{array}\right),$$
where the activation functions $\phi_{l,j,i}(\cdot )$ connect the i-th neuron of the l-th layer with the j-th neuron of the $\left(l+1\right)$-th layer, and $n_{l}$ is the number of neurons in the l-th layer.

Each activation function $\phi \left(u\right)$ is a combination of basis function b(u) and $spline\left(x\right)$ with the corresponding weights $w_{b}$ and $w_{s}$:$$\phi (u)=w_{b}b(u)+w_{s}\mathrm{spline}(u).$$

Here, the spline function $spline\left(u\right)$ is a linear combination of B-splines [21]:(3)$$\mathrm{spline}(u)=\sum_{k}c_{k}B_{k}(u).$$

The linear-KAN block in our work is a subnetwork described by Equation (1). Additionally, in Section 3.1, we show that adding BatchNorm layers can improve the performance of KAN-based networks. Therefore, the linear-KAN block can also include the indicated layers. We implemented the linear-KAN block using the optimized code from the efficient KAN project [34], which significantly accelerates the original pyKAN implementation [35], ensuring faster and more efficient processing.

#### 2.1.2. Conv-KAN Block

The convolutions used in classical CNNs can be described by the following equations:$$1D:{\left(K^{conv}*T\right)}_{i}=\sum_{l=-A}^{A}K_{i}^{conv}\cdot T_{i+l};$$
$$2D:{\left(K^{conv}*T\right)}_{i,j}=\sum_{m=-B}^{B}\sum_{l=-A}^{A}K_{i,j}^{conv}\cdot T_{i+l,j+m};$$
$$3D:{\left(K^{conv}*T\right)}_{i,j,k}=\sum_{n=-C}^{C}\sum_{m=-B}^{B}\sum_{l=-A}^{A}K_{i,j,k}^{conv}\cdot T_{i+l,j+m,k+n},$$
where $K^{conv}$ is a convolutional kernel of the corresponding dimension, and T is a signal (image) to be filtered.

The main difference between KAN convolutions and the convolutions used in CNNs lies in the kernel. In CNNs, it is made of real weights whereas in convolutional KANs, each element of the kernel, ϕ, is a learnable non-linear function that utilizes B-splines. In a KAN convolution, the kernel slides over an image T and applies the corresponding activation function ϕ(⋅) to the elements from T, like in a classic convolution. Thus, KAN convolutions are defined as [36]
$$1D:{\left(K^{\mathrm{KAN}}*T\right)}_{i}=\sum_{l=-A}^{A}\phi_{l}(T_{i+l});$$
$$2D:{\left(K^{\mathrm{KAN}}*T\right)}_{i,j}=\sum_{m=-B}^{B}\sum_{l=-A}^{A}\phi_{l,m}(T_{i+l,j+m});$$
$$3D:{\left(K^{\mathrm{KAN}}*T\right)}_{i,j,k}=\sum_{n=-C}^{C}\sum_{m=-B}^{B}\sum_{l=-A}^{A}\phi_{l,m,n}(T_{i+l,j+m,k+n}),$$
where $K^{KAN}$ is a KAN kernel of the corresponding dimension.

The Conv-KAN block is designed to replace the convolutional layer in the feature extractor of traditional neural network architectures.

We used the implementation from the FastKAN-Conv project [37] based on the paper [38]. It uses Gaussian radial basis functions to approximate the B-spline basis from Equation (3), which is the bottleneck of KAN and efficient KAN:$$B_{i}(u)=\exp\left(-\frac{{\left(u-u_{i}\right)}^{2}}{2h^{2}}\right),$$
where $u_{i}$ is a center, and h is a width of the corresponding radial function.

#### 2.1.3. KAN-Transformer Block

The classical transformer block consists of attention and MLP blocks. We propose to replace both the attention and MLP blocks with their KAN counterparts. The MLP block was shown in the previous subsection, and the attention block is considered below.

The original Attention mechanism was presented in the paper [39], and it can be described as
$$attention(Q,K,V)=\mathrm{Softmax}\left(\frac{QK^{T}}{\sqrt{d_{k}}}\right)V,$$
where $\frac{1}{\sqrt{d_{k}}}$ is a scaling factor, and Q, K, V are computed as the following:$$Q=W_{Q}X;$$
$$K=W_{K}X;$$
$$V=W_{V}X.$$

We propose to replace matrices $W_{Q}$ $W_{K}$ and $W_{V}$ with KAN function matrices (2):$$Q=\Phi_{Q}(X);$$
$$K=\Phi_{K}(X);$$
$$V=\Phi_{V}(X).$$

We used KAN-GPT implementation [40] as it is a more efficient realization for transformer architectures.

### 2.2. Architectures

To demonstrate the effectiveness of the KAN blocks described above, we selected the following neural network architectures utilized in the literature for HSI classification:MLP;1DCNN [26];2DCNN [27];3DCNN [28];3DCNN [29] and its modification NM3DCNN [30];SSFTT [31].

The neural network architectures were chosen for the study to include the basic neural network approach (MLP) as well as the most widely used neural network approaches based on other architectures. We considered convolutional-network-based approaches (1DCNN [26], 2DCNN [27], 3DCNN [29], NM3DCNN [30]), and in particular, approaches that we consider spectral-only (1DCNN [26]), spatial-only (2DCNN [27]), and spatial–spectral information (3DCNN [28], 3DCNN [29], NM3DCNN [30]), as well as an approach based on the transformer neural network architecture (SSFTT [31]). In each architecture, we replaced the classification layers with linear-KAN blocks of equivalent size. Additionally, the layers within the attention mechanism were substituted with linear-KAN Blocks, while the convolutional layers were replaced by Conv-KAN Blocks. For all KAN layers, we experimentally selected PReLU as the activation function and set the grid size to two. The remaining KAN parameters were adopted from the original implementation [34] as follows:Spline order = 3;Scale noise = 0.1;Scale base = 1.0;Scale spline = 1.0;Grid eps = 0.02;Grid range = [−1, 1]).

#### 2.2.1. MLP and Linear KAN Block

Given the continued relevance of MLPs and their frequent use in various HSI classification modifications [41,42,43], a classic multilayer perceptron was selected as our baseline architecture.

In our experiments, we studied different variants of MLP by varying the number of hidden layers and their width. For comparison, we used the linear-KAN block (see Section 2.1.1) with the same configuration, in which we also varied the number of spline nodes G (grid size). It is advisable to choose this parameter, taking into account that the total number of network parameters increases with G, as follows [21]:$$O((G+k)\cdot \sum_{l=1}^{L-1}n_{l+1}\cdot n_{l}),$$
where L is the number of layers, n is the number of functions in the l-th layer, G is the grid size, and k is the spline order.

As mentioned earlier, to enhance classification performance, we also incorporated BatchNorm layers before the hidden and classification layers.

#### 2.2.2. 1DCNN

The 1DCNN architecture, introduced by Wei Hu et al. [26], leveraged one-dimensional convolutional layers to extract deep spectral features for HSI classification. This architecture included a one-dimensional convolutional layer, a max pooling layer, and two fully connected layers (see Figure A1a). It’s important to note that this neural network operated solely on spectral data without incorporating spatial relationships.

Our proposed modification, referred to as the 1DCNN KAN, enhances the original 1DCNN architecture by replacing the classification block—comprising hidden and classification fully connected layers—with a linear-KAN block. Additionally, the one-dimensional convolutional layer in the feature extractor block is substituted with a Conv-KAN one-dimensional layer. The linear-KAN block maintains similar input and output dimensions and consists of two hidden layers, each with 512 units. This block is preceded by a BatchNorm1D layer for improved performance (see Figure A1b).

#### 2.2.3. 2DCNN

The 2DCNN architecture, introduced by Konstantinos Makantasis et al. [27], utilized two-dimensional convolutional layers to extract deep spatial features from HSIs. The initial 2D convolutional layer featured a 3 × 3 window size, with the number of convolutional kernels set to three times the number of spectral bands. The subsequent 2D convolutional layer also employed a 3 × 3 window, with the number of kernels tripled compared to the preceding layer. The classification block in this architecture comprised two linear layers (see Figure A2a), with the hidden layer containing neurons equivalent to six times the number of spectral bands. We propose replacing the two-dimensional convolutional layers with their KAN-based counterparts and substituting the classification block with a Linear-KAN block featuring 128 neurons in its hidden layer (see Figure A2b).

#### 2.2.4. 3DCNN by Luo

The 3DCNN architecture, introduced by Yanan Luo et al. [28], combined 3D and 2D convolutional layers to extract deep spatial–spectral features. The architecture included an initial 3D convolutional layer with a 24 × 3 × 3 window, followed by a 2D convolutional layer with a 3 × 3 window, and concluded with two linear layers in the classifier (see Figure A3a).

In our approach, we propose replacing the convolutional layers with their KAN-based counterparts, using fewer convolutional kernels. Specifically, the original architecture employed 90 and 64 kernels for the 3D and 2D convolutional layers, respectively, while the KAN modification reduced these to 64 and 32 kernels. Additionally, we replaced the classification block as previously described (see Figure A3b).

#### 2.2.5. 3DCNN by He

The 3DCNN He architecture, introduced by Mingyi He et al. [29], focused on using three-dimensional convolutions to extract spectral–spatial features, complemented by blocks of parallel one-dimensional convolutions to enhance feature diversity. The architecture begins with a 3D convolutional layer for extracting primary spectral–spatial features. This was followed by two ConvBlocks, each containing four parallel convolutional layers with varying kernel sizes to capture a broader range of spectral-spatial features (see Figure A4a). The design concluded with an additional 3D convolutional layer and a fully connected classification layer (see Figure A5a).

The proposed modification, referred to as 3DCNN He KAN (illustrated in Figure A5b), involves replacing all convolutional layers in the original 3DCNN He architecture with Conv-KAN blocks. Additionally, the ConvBlocks (see Figure A4a) are substituted with KAN-ConvBlocks (see Figure A4b), and the fully connected classification layer is replaced by a linear-KAN block, maintaining identical input and output dimensions. The modified classification block includes a single linear-KAN layer of the same size as in the original architecture, with a BatchNorm layer preceding it for enhanced performance.

#### 2.2.6. NM3DCNN

The NM3DCNN architecture [30], introduced by Firsov et al., represented an evolution of the earlier 3DCNN He architecture. This modification (illustrated in Figure A6a and Figure A7a) incorporated a BatchNorm layer following each convolutional layer to stabilize training and improve model performance while removing the Dropout layer from the original design.

The proposed modification to the NM3DCNN architecture, referred to as NM3DCNN KAN, involves several key updates. Specifically, all three-dimensional convolutional layers are replaced with Conv-KAN blocks. Additionally, the BNConvBlocks (see Figure A7a) are substituted with KAN-BNConvBlocks (see Figure A7b), and the fully connected classification layer is replaced by a linear-KAN block that maintains the same input and output dimensions, with one layer matching the size of the original version and preceded by a BatchNorm1D layer (see Figure A7b). Furthermore, the number of kernels in all convolutional layers has been reduced by half.

#### 2.2.7. SSFTT

The SSFTT transformer architecture [31], introduced by Le Sun et al., integrated a spatial–spectral feature extractor, a tokenizer, a transformer encoder, and a classification linear layer. This architecture was notable for its use of convolutional layers to extract spatial–spectral features and a trainable tokenizer with an attention mechanism (see Figure A8a).

Our proposed modification, referred to as SSFTT KAN (see Figure A8b), includes the following changes: The replacement of convolutional layers, i.e., substituting the convolutional layers in the feature extractor with Conv-KAN blocks while preserving the original input and output dimensions.

Modification of the attention block: Replacing the two fully connected layers within the attention block with a linear-KAN block. This block features two layers of the same size as the original, maintaining the input and output dimensions (based on the KAN GPT implementation [40]).

Modification of the encoder: Replacing the two fully connected linear layers within the encoder with a linear-KAN block. This block contains two layers of the same size as in the original version, preserving the input and output dimensions of the MLP block (following the KAN GPT implementation [40], which is tailored for transformer architectures).

Replacement of the classification block: Substituting the classification block with a linear-KAN block that includes a single linear-KAN layer of the same size as the original.

### 2.3. Description of Datasets

In our study, we utilized seven distinct open hyperspectral datasets, as detailed in Table 1, along with their respective characteristics. For the Pavia University and Indian Pines datasets, we randomly selected 20% of the pixels from each class to form the training set. The remaining datasets—PaviaC, Salinas, Houston 13, Houston 18, and KSC—had 10% of the pixels from each class randomly chosen for training. Consequently, the test sets comprised 80% of the data for PaviaU and Indian Pines and 90% for the other datasets. This data partitioning strategy was designed to optimize the architecture of a neural network classifier capable of performing effectively with a limited amount of training data. For the SSFTT architecture, PCA was applied to reduce the dimensionality to 30 components, consistent with the approach outlined in the original paper.

## 3. Results

### 3.1. Experimental Results

#### 3.1.1. MLP vs. KAN

The first phase of our experiments was to compare MLP and KAN in different variations. This stage can be divided into four experiments, where all models were trained from scratch. The training hyperparameters, such as learning rate and scheduler parameters, were chosen based on the training graphs for each classifier separately.

For the evaluation, we used the average accuracy value across all datasets and the average accuracy gain relative to MLP. This evaluation avoids taking into account the specifics of each individual dataset, since they differ in characteristics such as the number of channels, target classes, etc.

In the first experiment, we compared MLP and KAN without hidden layers, consisting only of an input layer (corresponding to the number of HSI channels) and an output layer (corresponding to the number of classes). Moreover, in this experiment, we varied the parameter G (Grid Size) in KAN, which is responsible for the number of nodes in the spline, in the range from 2 to 20. The accuracy results obtained are shown in Table 2 and Figure 1.

In the second experiment, we conducted a comparative analysis of MLP and KAN with varying numbers of neurons in the hidden layer (4, 8, 16, 32, 64, and 128 neurons). Additionally, we evaluated the impact of different values of the parameter G (2, 5, and 8).

These values of the parameter G were chosen based on the results obtained in the first experiment. In particular, we chose the value of G = 8 corresponding to the global maximum of the average classification accuracy. In addition, we took the values corresponding to two local maxima with smaller values of G = 2 and G = 5, since corresponding KAN networks have fewer parameters.

The results are presented in Table 3 and Figure 2.

In the third experiment, the performance of MLP and KAN models with two hidden layers was evaluated by varying their widths as in the second experiment (see Table 4 and Figure 3). Furthermore, we investigated the impact of the parameter G with two values (2 and 5). Such relatively small grid size values were chosen based on the growth in the number of network parameters.

In the fourth experiment, we investigated the effectiveness of applying batch normalization for MLP and KAN models. In particular, we considered the networks with two hidden layers of varying widths (16, 32, 64, 128). The value of parameter G was fixed at 2 and remained constant throughout the experiment. Batch normalization layers were applied before the hidden and output layers. Table 5 and Figure 4 show the results obtained.

#### 3.1.2. Partial Replacement with KAN-Blocks

In the second phase of our experiments, the impact of replacing individual blocks—specifically, the feature extractor block and the classifier block—was investigated to assess the significance of these architectural modifications. We chose the 1DCNN and 3DCNN He architectures for this analysis to compare architectures with markedly different depths. The training was conducted as described previously, with all models trained from scratch until accuracy and loss metrics stabilized. The results are detailed in Table 6 and Table 7 and Figure 5. In these tables, ‘vanilla’ refers to the original architecture, ‘KAN-FE’ indicates the replacement of the feature extractor block with KAN, ‘KAN-head’ denotes the replacement of the classifier block with KAN, and ‘Full KAN’ represents the replacement of both blocks. The final column shows the average performance gain relative to the vanilla architecture.

#### 3.1.3. Convolutional and Transformer Networks

In the third phase of our experiments, we focused on evaluating the efficiency of fully replacing classical network layers with their KAN analogs in both convolutional and transformer architectures. This replacement was implemented across various network components, including the feature extractor blocks (convolutional layers), transformer attention mechanisms, and MLP blocks. Additionally, the linear layers in the classification block were substituted with corresponding KAN blocks.

Prior to model training, we standardized the data during the preprocessing stage. For both SSFTT and SSFTT KAN networks, we applied principal component analysis (PCA) to extract the top 30 principal components. All models were trained from scratch, continuing until both training accuracy and loss indicators plateaued. The learning rate was carefully selected within the range of [0.001, 0.4], and we utilized the Adam optimizer in conjunction with the cross-entropy loss function. Scheduler parameters were adjusted based on convergence, with hyperparameter tuning for KAN networks being similar to classical architectures, though KAN networks exhibited faster convergence.

The results of these experiments are detailed in Figure 6 and Table 8 and Table 9. We organized the results according to the convolution type and window size employed by the networks for feature extraction. Specifically, the 1DCNN and 1DCNN KAN architectures rely solely on spectral features without leveraging spatial context. In contrast, 2DCNN and 2DCNN KAN utilize 2D convolutions to process spatial data. The 3DCNN Luo and 3DCNN Luo KAN models incorporate spectral–spatial convolutions using a 3 × 3 spatial window, while 3DCNN He, 3DCNN He KAN, NM3DCNN, and NM3DCNN KAN employ a larger 7 × 7 spatial window. The SSFTT and SSFTT KAN models stand out, utilizing a 13 × 13 spatial patch size. As with previous analyses, the last two columns of the tables present the classification quality and gain, averaged across all datasets for each pair of original and KAN-based networks.

## 4. Discussion

All the solutions evaluated delivered a high level of quality. Let us begin by examining classical architectures.

The initial stage of experiments (Table 2, Table 3, Table 4 and Table 5) indicated that KANs exhibited superior classification quality metrics in comparison to conventional MLPs. We did not identify an obvious dependence of the classification accuracy on the G parameter. For this reason, we gave preference to KAN networks with small values of the G parameter (2, 5) since they had fewer parameters. On the other hand, increasing the number and size of layers led to a slight increase in the classification quality. For instance, the maximal average accuracy for KAN without hidden layers was 89.95%. The KAN with one hidden layer achieved 91.53%, while with two hidden layers, it attained 92.05%. It is worth noting that an increase in the number of neurons in the hidden layers resulted in a reduction in the average accuracy gain relative to MLP since the accuracy of MLP increased. During the experiments, it was found that utilizing batch normalization before hidden layers, which stabilizes training and improves the generalization ability of classifiers, allows achieving an accuracy of 93.91%.

The results from the second phase of experiments (Table 6 and Table 7) indicate that replacing traditional blocks with their KAN analogs consistently enhances classification quality. Specifically, the replacement of both the feature extractor block and the classification block yields the best results, with improvements of 2.91% for 1D-CNN and 5.06% for 3D-CNN He. For 1D-CNN, substituting either the feature extractor block or the classifier block results in average quality increases of 2.24% and 2.41%, respectively. In contrast, for 3D-CNN He, replacing only the classifier block produces a more substantial improvement—3.72%—compared to replacing only the feature extractor block, which results in a 0.68% increase. Thus, it can be concluded that for deeper networks, both blocks should be replaced, while for shallower networks, replacing just one of them is sufficient.

In the third phase of our experiments (Table 8 and Table 9), we studied traditional CNN and transformer architectures with their full KAN analogs. Among the traditional convolutional networks, the 1D-CNN, which relies solely on spectral information, yielded the weakest average performance. In contrast, the 2D-CNN, which incorporates spatial feature convolution, outperformed the 1D-CNN by an average of 1.19%. It consistently demonstrated equal to or slightly superior results across all examined scenes. This highlights the significant advantage of integrating local spatial context in the analysis of hyperspectral remote sensing images. Among the networks utilizing spectral–spatial convolutions, such as 3D-CNN Luo, 3D-CNN He, and NM3DCNN, we find NM3DCNN to be the most effective. It consistently delivered superior results across nearly all analyzed scenes, with the exceptions of Houston13 and KSC. Notably, NM3DCNN demonstrated a substantial improvement in quality for the Indian Pines scene, showing a 5.75% enhancement over 3D-CNN He and an 8.3% improvement over 3D-CNN Luo. This network also outperformed those relying solely on spectral or spatial convolutions (1D-CNN and 2D-CNN).

The SSFTT network achieved markedly better results than the other classical neural networks evaluated, with the most significant differences observed for the Indian Pines and KSC scenes, showing improvements of 8.58% and 8.84% over NM3DCNN, respectively. We attribute this improvement to the significantly larger spatial window (context) size and attention mechanism used in SSFTT.

While the primary goal of this study was not to compare various neural network architectures but rather to showcase the benefits of KAN-based architectures, the findings are noteworthy. Replacing traditional layers with KAN analogs proved beneficial, with improvements in quality observed across all cases. For CNNs, the average enhancement ranged from 2.44% to 5.05%, with the most substantial effect achieved by the 3D-CNN He network. Although the transformer-based SSFTT KAN network exhibited a more modest average increase of less than 1%, it nonetheless achieved the highest classification quality.

The primary trends observed in classical architectures are also evident in their KAN counterparts. For instance, 2D-CNN KAN outperforms 1D-CNN KAN by an average of 0.72%, NM3DCNN KAN surpasses 2D-CNN KAN by 2.15%, and SSFTT KAN is 2% better than NM3DCNN KAN. The highest classification accuracy of 99.2% is achieved by the SSFTT KAN transformer network. Notably, the f-measure results generally support these conclusions based on classification accuracy.

Thus, we considered the application of KAN-based neural networks in one narrow domain, namely, pixel-wise classification of hyperspectral data. We showed that the use of KAN blocks in modern neural network architectures allowed us to improve the quality of the solution compared to classical architectures. This confirms the provisions of the seminal paper [21].

The main difference in KAN layers is the use of univariate weight functions parameterized as a spline instead of scalar weights of classical layers. We believe that this feature allowed KAN analogs to outperform classical networks in our experiments. This feature allowed us, albeit to varying degrees, to improve the results not only in classical fully-connected layers but also in convolutional layers and the attention mechanism. We believe that further research on KAN networks will improve the results both in other domains and using different network architectures.

## 5. Conclusions

This paper applies KAN networks to the pixel-wise classification of HSIs, starting by replacing the traditional MLP with the recently proposed KAN network. The results highlight its advantages, particularly when batch normalization layers are incorporated, yielding the best outcomes. KAN equivalents were then introduced for six neural network architectures commonly used in HSI classification: 1D (1D-CNN), 2D (2D-CNN), and 3D convolutional networks (3D-CNN Luo, 3D-CNN He, NM3DCNN), along with a transformer model (SSFTT). Both partial and full replacements of classical convolutional and classification layers, as well as attention mechanisms, with KAN analogs, are proposed.

By leveraging seven publicly available hyperspectral datasets, we demonstrated that the modified KAN architectures consistently outperformed traditional neural networks in all evaluated cases. Replacing the conventional MLP with KAN resulted in higher classification quality for both shallow and deep networks. Incorporating batch normalization within the hidden layers of KAN further enhanced classification performance. Fully substituting all layers in neural networks with their KAN counterparts yielded better results than replacing individual blocks alone. Among all the architectures examined, the transformer-based SSFTT KAN network achieved the best performance.

## Figures and Tables

**Figure 1 sensors-24-07683-f001:**
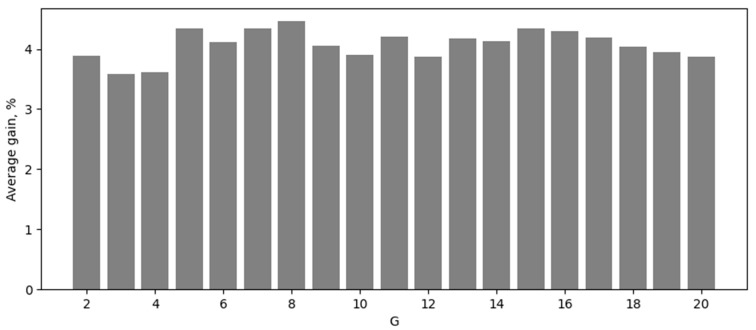
Average accuracy gain for neural network models without hidden layer, in percentage.

**Figure 2 sensors-24-07683-f002:**
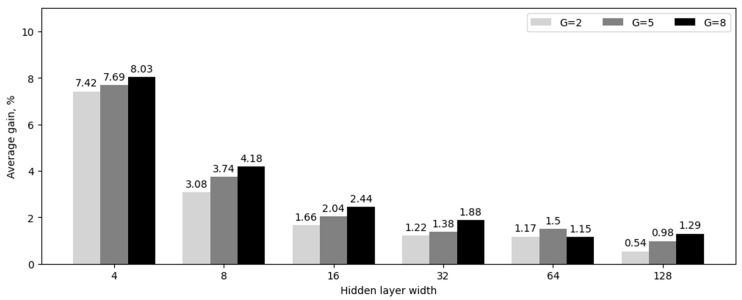
Average accuracy gain for neural network models with one hidden layer, in percent.

**Figure 3 sensors-24-07683-f003:**
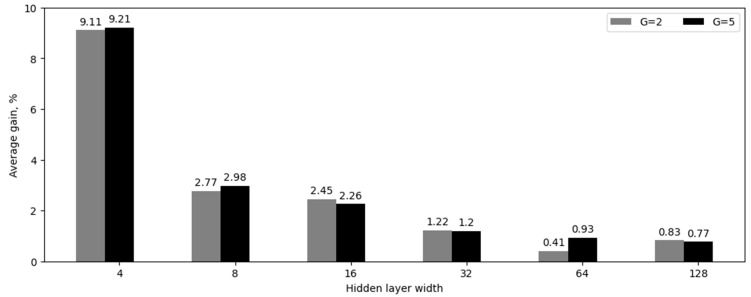
Average accuracy gain for neural network models with two hidden layers, in percentage.

**Figure 4 sensors-24-07683-f004:**
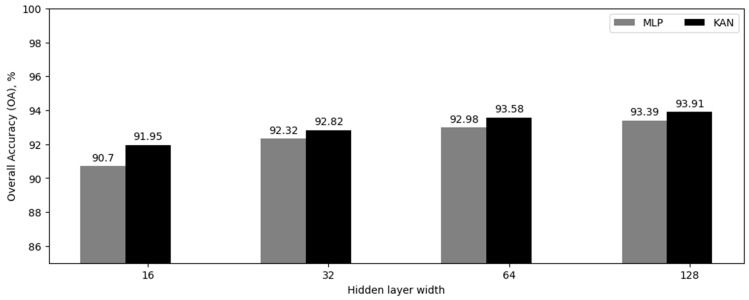
Overall classification accuracy (OA) for neural network models with batch normalization, in percentage.

**Figure 5 sensors-24-07683-f005:**
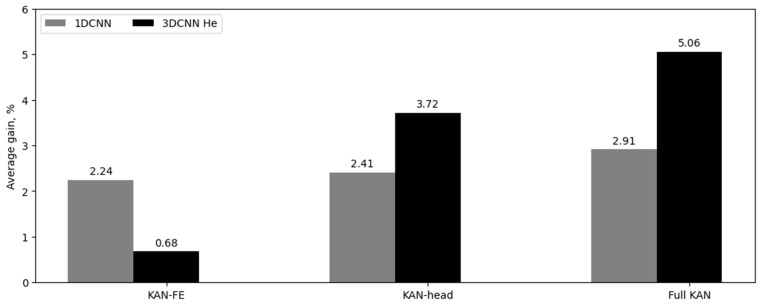
Average accuracy gain for 1DCNN and 3DCNN He, in percentage.

**Figure 6 sensors-24-07683-f006:**
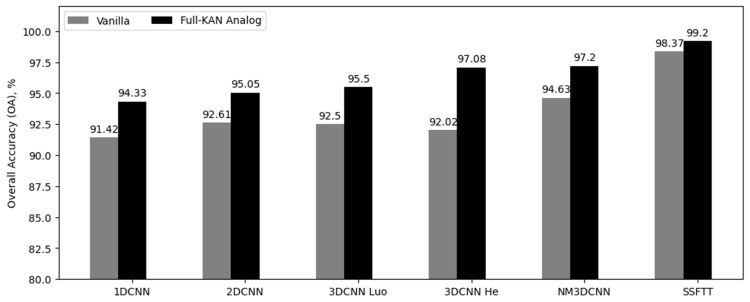
Overall classification accuracy (OA) for various neural network models and datasets, in percentage.

**Table 1 sensors-24-07683-t001:** Datasets.

Dataset Name	Size	Spectral Channels	Number of Classes	Classes
PaviaU	610 × 340	103	9	Void, Asphalt, Meadows, Gravel, Trees, Painted metal sheets, Bare soil, Bitumen, Self-blocking bricks, Shadows
PaviaC	1096 × 715	102	9	Void, Water, Trees, Meadows, Self-blocking bricks, Bare soil, Asphalt, Bitumen, Tiles, Shadows
Salinas	512 × 217	204	16	Void, Green weeds 1, Green weeds 2, Fallow, Fallow rough plow, Fallow smooth, Stubble, Celery, Grapes Untrained, Soil vineyard, Corn senesced, Lettuce romaine 4 week, Lettuce romaine 5 week, Lettuce romaine 6 week, Lettuce romaine 7 week, Vineyard_untrained, Vineyard_vertical
Indian Pines	145 × 145	200	16	Void, Alfalfa, Corn-notill, Corn-min, Corn, Grass/Pasture, Grass/Trees, Grass pasture/mowed, Hay-windrowed, Oats, Soybeans-no till, Soybeans-min till, Soybeans-clean till, Wheat, Woods, Buildings-grass-trees-Drive, Store-steel-towers
Houston 13	954 × 210	48	7	Void, Grass healthy, Grass stressed, Trees, Water, Residential Buildings, Non-residential Buildings, Road
Houston 18	954 × 210	48	7	Void, Grass healthy, Grass stressed, Trees, Water, Residential Buildings, Non-residential Buildings, Road
KSC	512 × 614	176	13	Void, Scrub, Willow swamp, CP hammock, Slash pine, Oak/Broadleaf, Hardwood, Swamp, Graminoid marsh, Spartina marsh, Cattail marsh, Salt marsh, Mud flats, Water

**Table 2 sensors-24-07683-t002:** Overall classification accuracy (OA) for neural network models without hidden layer, in percentage.

Model Name	Dataset Name								
	PaviaU *	PaviaC	Salinas	Indian Pines *	Houston13	Houston 18	KSC	Average	Average Gain
KAN (in, out) G20	92.29	98.26	91.91	76.79	94.15	84.47	87.75	89.37	3.88
KAN (in, out) G19	92.30	98.26	91.96	76.79	94.36	84.43	88.04	89.44	3.95
KAN (in, out) G18	92.49	98.26	91.95	77.03	94.19	84.85	87.98	89.53	4.04
KAN (in, out) G17	92.47	98.26	91.99	79.08	93.93	84.38	87.67	89.68	4.19
KAN (in, out) G16	92.57	98.28	92.06	79.17	94.32	85.14	87.01	89.79	4.3
KAN (in, out) G15	92.60	98.28	92.14	78.95	94.28	85.63	86.94	89.83	4.34
KAN (in, out) G14	92.73	98.27	92.19	76.61	94.19	85.36	88.04	89.62	4.13
KAN (in, out) G13	92.64	98.28	92.13	77.04	94.02	85.77	87.75	89.66	4.17
KAN (in, out) G12	91.97	98.13	91.54	78.61	93.41	85.27	86.68	89.37	3.88
KAN (in, out) G11	92.77	98.28	92.19	78.29	93.63	85.65	87.01	89.68	4.2
KAN (in, out) G10	92.05	98.12	91.50	78.08	93.37	85.41	87.22	89.39	3.9
KAN (in, out) G9	92.95	98.14	91.53	77.80	94.11	85.89	86.36	89.54	4.05
KAN (in, out) G8	93.27	98.49	92.00	78.24	94.15	86.23	87.27	**89.95**	**4.46**
KAN (in, out) G7	93.35	98.50	91.98	77.31	94.11	86.25	87.33	89.83	4.34
KAN (in, out) G6	91.93	98.17	91.14	77.99	94.15	86.39	87.54	89.61	4.12
KAN (in, out) G5	91.71	98.41	91.95	77.49	95.01	87.14	87.20	89.84	4.35
KAN (in, out) G4	91.53	98.16	90.99	77.01	93.24	85.99	86.78	89.1	3.61
KAN (in, out) G3	91.13	98.12	90.81	77.05	93.33	85.91	87.16	89.07	3.58
KAN (in, out) G2	90.62	98.30	90.87	78.64	93.85	86.36	87.08	89.38	3.89
*MLP (in, out)*	*87.15*	*97.87*	*90.35*	*75.64*	*88.17*	*80.86*	*78.41*	*85.49*	*-*

* The training sample size for PaviaU and Indian Pines consisted of 20% samples per class, while for other datasets, it was 10% per class. The line in italics and gray serves as the basis for comparison. The best values for average accuracy and gain are in bold.

**Table 3 sensors-24-07683-t003:** Overall classification accuracy (OA) for neural network models with one hidden layer, in percentage.

Model Name	Dataset Name								
	PaviaU *	PaviaC	Salinas	Indian Pines *	Houston13	Houston 18	KSC	Average	Average Gain
KAN (in, 4, out) G8	91.07	97.25	89.35	72.30	92.72	87.36	85.86	**87.99**	**8.03**
KAN (in, 4, out) G5	92.05	98.02	88.76	72.36	92.50	87.63	82.26	87.65	7.69
KAN (in, 4, out) G2	91.23	97.38	89.23	73.17	91.03	86.46	83.14	87.38	7.42
*MLP (in, 4, out)*	*88.24*	*97.07*	*80.13*	*64.57*	*81.50*	*84.93*	*63.28*	*79.96*	*-*
KAN (in, 8, out) G8	93.14	98.18	90.28	75.53	95.40	88.95	88.89	**90.05**	**4.18**
KAN (in, 8, out) G5	92.05	98.67	90.42	75.29	95.06	89.54	86.24	89.61	3.74
KAN (in, 8, out) G2	91.98	98.20	91.32	76.24	89.90	89.67	85.37	88.95	3.08
*MLP (in, 8, out)*	*91.14*	*97.94*	*85.69*	*72.39*	*89.45*	*86.31*	*78.20*	*85.87*	*-*
KAN (in, 16, out) G8	93.42	98.23	91.60	79.75	94.15	89.96	88.36	**90.78**	**2.44**
KAN (in, 16, out) G5	93.36	98.67	91.50	79.36	93.98	90.49	85.29	90.38	2.04
KAN (in, 16, out) G2	93.18	98.45	91.55	78.81	92.55	90.29	85.16	90.00	1.66
*MLP (in, 16, out)*	*93.04*	*98.15*	*91.02*	*78.73*	*88.13*	*87.68*	*81.64*	*88.34*	*-*
KAN (in, 32, out) G8	94.63	98.49	92.06	80.92	94.67	90.22	88.57	**91.37**	**1.88**
KAN (in, 32, out) G5	94.88	98.67	91.74	80.84	93.50	90.70	85.73	90.87	1.38
KAN (in, 32, out) G2	93.90	98.49	92.02	80.77	91.94	89.61	88.26	90.71	1.22
*MLP (in, 32, out)*	*93.88*	*98.16*	*91.67*	*80.27*	*91.81*	*88.17*	*82.47*	*89.49*	*-*
KAN (in, 64, out) G8	94.38	98.44	91.25	81.33	93.63	90.72	88.59	91.19	1.15
KAN (in, 64, out) G5	94.49	98.53	92.12	81.44	94.86	90.51	88.81	**91.53**	**1.50**
KAN (in, 64, out) G2	94.04	98.48	92.20	82.12	92.78	89.52	89.34	91.21	1.17
*MLP (in, 64, out)*	*93.71*	*98.15*	*91.77*	*81.16*	*92.11*	*88.28*	*85.08*	*90.03*	*-*
KAN (in, 128, out) G8	94.44	98.34	91.96	81.51	94.88	90.33	88.24	**91.38**	**1.29**
KAN (in, 128, out) G5	93.82	98.57	92.00	81.32	94.15	90.36	87.33	91.07	0.98
KAN (in, 128, out) G2	93.48	98.39	92.02	80.34	92.77	89.89	87.56	90.63	0.54
*MLP (in, 128, out)*	*93.18*	*98.16*	*91.20*	*80.27*	*92.59*	*88.28*	*86.97*	*90.09*	*-*

* The training sample size for PaviaU and Indian Pines consisted of 20% samples per class, while for other datasets, it was 10% per class. The lines in italics and gray serve as the basis for comparison. The best values for average accuracy and gain are in bold.

**Table 4 sensors-24-07683-t004:** Overall classification accuracy (OA) for neural network models with two hidden layers, in percentage.

Model Name	Dataset Name								
	PaviaU *	PaviaC	Salinas	Indian Pines *	Houston13	Houston 18	KSC	AVG	Gain
KAN (in, 4, 4, out) G5	93.42	97.93	90.74	72.80	95.06	89.05	80.02	**88.43**	**9.21**
KAN (in, 4, 4, out) G2	92.36	97.84	90.21	72.74	93.72	88.57	82.89	88.33	9.11
*MLP (in, 4, 4, out)*	*89.79*	*96.78*	*78.94*	*60.91*	*72.75*	*87.78*	*67.64*	*79.22*	*-*
KAN (in, 8, 8, out) G5	94.86	98.45	91.62	76.58	93.67	89.98	85.12	**90.04**	**2.98**
KAN (in, 8, 8, out) G2	94.06	97.98	91.45	74.89	95.23	90.03	85.23	89.83	2.77
*MLP (in, 8, 8, out)*	*93.43*	*98.06*	*90.56*	*75.13*	*85.66*	*89.41*	*77.21*	*87.06*	*-*
KAN (in, 16, 16, out) G5	94.71	98.35	92.45	79.99	94.67	90.63	85.41	90.88	2.26
KAN (in, 16, 16, out) G2	95.02	98.22	91.65	80.31	95.92	90.09	86.32	**91.07**	**2.45**
*MLP (in, 16, 16, out)*	*92.68*	*97.12*	*91.41*	*75.40*	*95.66*	*86.12*	*82.01*	*88.62*	*-*
KAN (in, 32, 32, out) G5	94.43	98.45	92.13	82.24	94.93	91.08	86.97	91.46	1.2
KAN (in, 32, 32, out) G2	94.88	98.69	91.88	82.02	94.80	90.89	87.25	**91.48**	**1.22**
*MLP (in, 32, 32, out)*	*94.54*	*98.21*	*91.58*	*78.12*	*96.75*	*87.81*	*84.87*	*90.26*	*-*
KAN (in, 64, 64, out) G5	95.50	98.50	92.95	82.98	95.32	91.36	87.22	**91.97**	**0.93**
KAN (in, 64, 64, out) G2	95.00	98.62	91.93	81.15	94.02	90.99	88.47	91.45	0.41
*MLP (in, 64, 64, out)*	*95.43*	*98.27*	*91.78*	*82.62*	*93.07*	*90.36*	*85.79*	*91.04*	*-*
KAN (in, 128, 128, out) G5	95.31	98.66	92.80	82.71	95.79	91.10	87.60	91.99	0.77
KAN (in, 128, 128, out) G2	94.84	98.34	92.05	83.79	96.36	90.98	88.00	**92.05**	**0.83**
*MLP (in, 128, 128, out)*	*94.66*	*97.72*	*90.07*	*83.47*	*95.01*	*90.83*	*86.78*	*90.79*	*-*

* The training sample size for PaviaU and Indian Pines consisted of 20% samples per class, while for other datasets, it was 10% per class. The lines in italics and gray serve as the basis for comparison. The best values for average accuracy and gain are in bold.

**Table 5 sensors-24-07683-t005:** Overall classification accuracy (OA) for neural network models with batch normalization, in percentage.

Model Name	Dataset Name								
	PaviaU *	PaviaC	Salinas	Indian Pines *	Houston13	Houston 18	KSC	Average	Average Gain
KAN (in, 16, 16, out) G2	95.81	98.66	93.56	80.63	94.84	89.50	90.70	**91.95**	**1.25**
*MLP (in, 16, 16, out)*	*92.85*	*97.91*	*91.16*	*80.28*	*97.09*	*87.25*	*88.38*	*90.70*	
KAN (in, 32, 32, out) G2	95.12	98.61	94.26	83.32	96.96	90.34	91.14	**92.82**	0.5
*MLP (in, 32, 32, out)*	*94.49*	*98.29*	*92.66*	*84.08*	*97.27*	*89.46*	*90.02*	*92.32*	
KAN (in, 64, 64, out) G2	96.14	98.98	93.70	**87.86**	96.53	91.24	90.61	**93.58**	0.6
*MLP (in, 64, 64, out)*	*95.69*	*98.47*	*92.98*	*85.77*	*97.01*	*89.92*	*91.03*	*92.98*	
KAN (in, 128, 128, out) G2	**96.34**	**99.03**	**94.22**	**87.01**	**97.57**	**92.04**	91.22	**93.91**	0.52
*MLP (in, 128, 128, out)*	*95.92*	*98.65*	*93.53*	*86.35*	*97.53*	*90.07*	** *91.73* **	*93.39*	

* The training sample size for PaviaU and Indian Pines consisted of 20% samples per class, while for other datasets, it was 10% per class. The lines in italics and gray serve as the basis for comparison. The best are in bold.

**Table 6 sensors-24-07683-t006:** Overall classification accuracy (OA) for 1DCNN and datasets, in percentage.

Model Name	Dataset Name							Average	Average Gain
	PaviaU *	PaviaC	Salinas	Indian Pines *	Houston13	Houston 18	KSC		
KAN-FE	95.37	98.96	94.70	86.87	97.22	92.91	89.64	93.66	2.24
KAN-head	95.30	98.85	94.41	87.10	98.87	92.42	89.90	93.83	2.41
Full KAN	**95.68**	**99.07**	**95.28**	**88.90**	**96.88**	**93.63**	**90.91**	**94.33**	**2.91**
*Vanilla*	*95.17*	*98.20*	*91.93*	*85.82*	*92.63*	*91.32*	*84.87*	*91.42*	*-*

* The training sample size for PaviaU and Indian Pines consisted of 20% samples per class, while for other datasets, it was 10% per class. The line in italics and gray serves as the basis for comparison. The best values are in bold.

**Table 7 sensors-24-07683-t007:** Overall classification accuracy (OA) for 3DCNN He and datasets, in percentage.

Model Name	Dataset Name							Average	Average Gain
	PaviaU *	PaviaC	Salinas	Indian Pines *	Houston13	Houston 18	KSC		
KAN-FE	98.74	99.45	92.25	89.39	91.69	90.62	86.77	92.70	0.68
KAN-head	98.71	99.34	96.65	92.77	95.15	95.25	92.35	95.74	3.72
Full KAN	**98.76**	**99.71**	**98.14**	**96.47**	**97.04**	**95.82**	**93.66**	**97.08**	**5.06**
*Vanilla*	*98.05*	*97.39*	*93.03*	*84.45*	*90.40*	*91.98*	*88.87*	*92.02*	*-*

* The training sample size for PaviaU and Indian Pines consisted of 20% samples per class, while for other datasets, it was 10% per class. The line in italics and gray serves as the basis for comparison. The best values are in bold.

**Table 8 sensors-24-07683-t008:** Overall classification accuracy (OA) for various neural network models and datasets, in percentage.

Model Name	Dataset Name							Average	Average Gain
	PaviaU *	PaviaC	Salinas	Indian Pines *	Houston13	Houston 18	KSC		
Convolutions of spectral features									
1DCNN	95.17	98.20	91.93	85.82	92.63	91.32	84.87	91.42	2.92
1DCNN KAN	**95.68**	**99.07**	**95.28**	**88.90**	**96.88**	**93.63**	**90.91**	**94.33**	
Convolutions of spatial features (window size 3 × 3)									
2DCNN	97.93	99.03	93.10	86.56	93.78	93.19	84.72	92.61	2.44
2DCNN KAN	**99.12**	**99.57**	**96.52**	**93.77**	**95.97**	**94.26**	**86.19**	**95.05**	
Convolutions of spectral–spatial features (window size 3 × 3)									
3DCNN Luo	96.92	99.21	93.99	81.90	95.69	92.45	87.37	92.50	3.00
3DCNN Luo KAN	**99.02**	**99.57**	**96.97**	**91.33**	**96.77**	**94.16**	**90.72**	**95.50**	
Convolutions of spectral–spatial features (window size 7 × 7)									
3DCNN He	98.05	97.39	93.03	84.45	90.40	91.98	88.87	92.02	5.05
3DCNN He KAN	**98.76**	**99.71**	**98.14**	**96.47**	**97.04**	**95.82**	**93.66**	**97.08**	
NM3DCNN	99.33	99.57	96.78	90.20	94.41	95.53	86.61	94.63	2.57
NM3DCNN KAN	**99.52**	**99.75**	**98.01**	**95.39**	**97.53**	**95.84**	**94.40**	**97.20**	
Convolutions of spectral–spatial features (window size 13 × 13), 30 principal components									
SSFTT	99.86	99.88	99.85	98.78	98.57	96.22	95.45	98.37	0.8271
SSFTT KAN	**99.92**	**99.93**	**99.97**	**99.24**	**99.46**	**97.12**	**98.76**	**99.20**	

* The training sample size for PaviaU and Indian Pines consisted of 20% samples per class, while for other datasets, it was 10% per class.

**Table 9 sensors-24-07683-t009:** Weighted F1 measure for various neural network models and datasets.

Model Name	Dataset Name							Average	Average Gain
	PaviaU *	PaviaC	Salinas	Indian Pines *	Houston13	Houston 18	KSC		
Convolutions of spectral features									
1DCNN	0.9524	0.9811	0.9165	0.8577	0.9262	0.9125	0.8469	0.9133	0.0295
1DCNN KAN	**0.9566**	**0.9903**	**0.9525**	**0.8883**	**0.9680**	**0.9356**	**0.9088**	**0.9428**	
Convolutions of spatial features (window size 3 × 3)									
2DCNN	0.9788	0.9898	0.9307	0.8640	0.9366	0.9311	0.8465	0,9253	0.2442
2DCNN KAN	**0.9910**	**0.9951**	**0.9646**	**0.9353**	**0.9581**	**0.9423**	**0.8606**	**0,9495**	
Convolutions of spectral–spatial features (window size 3 × 3)									
3DCNN Luo	0.9689	0.9914	0.9392	0.8185	0.9563	0.9243	0.8728	0.9244	0.0300
3DCNN Luo KAN	**0.9898**	**0.9949**	**0.9690**	**0.9130**	**0.9672**	**0.9413**	**0.9067**	**0.9545**	
Convolutions of spectral–spatial features (window size 7 × 7)									
3DCNN He	0.9801	0.9737	0.9301	0.8442	0.9036	0.9195	0.8882	0.9199	0.0506
3DCNN He KAN	**0.9874**	**0.9968**	**0.9813**	**0.9646**	**0.9700**	**0.9577**	**0.9364**	**0.9706**	
NM3DCNN	0.9929	0.9954	0.9676	0.9014	0.9437	0.9552	0.8660	0.9460	0.0258
NM3DCNN KAN	**0.9951**	**0.9973**	**0.9800**	**0.9537**	**0.9751**	**0.9583**	**0.9438**	**0.9719**	
Convolutions of spectral–spatial features (window size 13 × 13), 30 principal components									
SSFTT	0.9985	0.9987	0.9985	0.9877	0.9856	0.9621	0.9543	0.9836	0.008271
SSFTT KAN	**0.9990**	**0.9992**	**0.9996**	**0.9923**	**0.9945**	**0.9712**	**0.9875**	**0.9919**	

* The training sample size for PaviaU and Indian Pines consisted of 20% samples per class, while for other datasets, it was 10% per class. The best values are in bold.

## Data Availability

The data presented in this study are available on request from the corresponding author.

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
