# Peer review of "HyperKAN: Kolmogorov–Arnold Networks Make Hyperspectral Image Classifiers Smarter"

_sensors, 2024, doi:10.3390/s24237683_

Round 1

Reviewer 1 Report

Comments and Suggestions for Authors

Please see attachments.

Author Response

Dear Reviewer,

Thank you very much for your careful reading of the article and your comments! We have taken all your comments into account in the updated version of the article.

We have highlighted all changes made to the article in green color.

The answers to your comments are given below.

Comments 1: Line 15, in abstract, you should briefly introduce what you have done, instead of using words like “suggest” which confusing the readers that you might have not completed the mentioned work.

Response 1: Thank you for your comment. We have modified the abstract according to your comment. In particular, we have replaced a phrase “suggest  replacing” with “replaced”. Other changes are shown in color.

Comments 2: Line 98-107. The reference format of Sensors doesn’t support you to use the sentence such as “[ref_num] have completed (....)”. Please rewrite them as “[Authors] completed (...) [ref_num]”. The paragraph of line 130-141 gives out a good example for appropriate of references.

Response 2: Thank you for your comment. We have carefully reviewed the entire text and corrected the citations in accordance with your recommendations.

Comments 3: Section 3.1 “datasets” should appear in section 2 as the materials of this study.

Response 3: Thank you for your comment. We have moved Section 3.1 to Section 2.3 according to your comment.

Comments 4: The description of section 2.2 is some kind of bulky (both for description and figures). I suggest reshaping those architecture figures into one or two figures, and emphasis description to distinguish each neutral architectures and illustrate “why these architectures are selected”.

Response 4: Thank you for your comment. We have moved the neural network architecture diagrams to Appendix A. This has significantly reduced the size of Section 2.2 without reducing the information content of the figures. We have also added a rationale for choosing each architecture at the beginning of Section 2.2 (see lines 211-217).

Comments 5: Section 3.2, explain the description of “average gain” to emphasize its importance as final comparison value.

Response 5: Thank you for your comment. We have added the following explanation at the beginning of Subsection 3.1.1 (see lines 349-352):

“For the evaluation, we used the average accuracy value across all datasets and the average accuracy gain relative to MLP. This evaluation avoids taking into account the specifics of each individual dataset, since they differ in characteristics such as the number of channels, target classes, etc.”

Comments 6: Line 359, explain why choose G2, 5 and 8 instead of other combinations.

Response 6: Thank you for your comment. We have added the following clarification before Table 2: (see lines 362-366):

These values of the parameter G were chosen based on the results obtained in the first experiment. In particular, we chose the value of G=8 corresponding to the global maximum of the average classification accuracy. In addition, we took the values corresponding to two local maxima with smaller values of G=2 and G=5, since corresponding KAN networks have fewer parameters.

Comments 7: I suggest you apply figures to all the tables in section 3, to highlight each peak.

Response 7: Thank you for your comment. We have added diagrams to almost all tables with experimental results. Please note that Figure 5 shows the results for both Table 6 and Table 7. We have not added a figure for Table 9 to avoid cluttering the paper (Figure 6 shows the diagram for the corresponding overall classification accuracy values).

Comments 8: For Table 2, highlight or distinguish the last line which plays the role of controlling group, instead of act them as another set of results. Same for the other tables.

Response 8: Thank you for your comment. We have highlighted in italics and gray those rows in the tables that serve as a basis for comparison (basic MLP classifiers or pure versions of modified networks).

Comments 9: The discussion part focus on results description instead of digging the origins of advantages from KAN analogs. I’m not sure whether the discussion parts of similar papers usually appears same as this manuscript.

Response 9: Thank you for your comment. In accordance with your comment, we have added a rationale to the end of section 4 (see lines 520-531). Briefly, we noted that KAN layers use univariate weight functions parameterized as a spline instead of scalar weights that allowed KAN analogs to outperform classical networks in our experiments.

Thank you again for your comments. We hope that the revised version of the article meets the high standards of the journal.

Reviewer 2 Report

Comments and Suggestions for Authors

In this article authors have presented their work on modifications to Kolmogorov-Arnold Networks. The modified methods are employed to study classifications of hyperspectral images obtained from three separate datasets. It is found that modified KAN methods show better classification accuracy compared to MLP based classification for all the datasets. This is very well written manuscript with conclusions presented with appropriate data.This manuscript could be published as it is (with corrections of minor spelling errors).

I have following specific comments:

1. If possible then two figures could be created to represent some data from Table 4 and Table 5, showing that increasing the width of hidden layers
(4, 16, 32, 64, 128) results in convergence of gain of modified KAN against MLP. That is for smaller widths KAN gives better results than corresponding MLP but this gain reduces for higher widths.

2. It would be useful to discuss briefly why only for Houston 18 datasets, MLP performs better for higher widths. For example Table 3, widths 64 and 128, it seems that MLP has better accuracy than some KAN based methods.

3. Spellings could be checked once again, for example line 77, convolved should be convoluted.

Overall, this is a very nicely written manuscript and could be accepted without any modifications.

Author Response

Dear Reviewer,

Thank you very much for your careful reading of the article and your comments! We have taken all your comments into account in the updated version of the article.

We have highlighted all changes made to the article in green color.

The answers to your comments are given below.

Comments 1: If possible then two figures could be created to represent some data from Table 4 and Table 5, showing that increasing the width of hidden layers (4, 16, 32, 64, 128) results in convergence of gain of modified KAN against MLP. That is for smaller widths KAN gives better results than corresponding MLP but this gain reduces for higher widths.

Response 1: Thank you very much for this observation. We have added diagrams to almost all tables with experimental results.

Figures 2 and 3 show a trend where the advantage of KAN networks decreases with increasing layer width.

Please note that Figure 5 shows the results for both Table 6 and Table 7. We have not added a figure for Table 9 to avoid cluttering the paper (Figure 6 shows the diagram for the corresponding overall classification accuracy values).

Comments 2: It would be useful to discuss briefly why only for Houston 18 datasets, MLP performs better for higher widths. For example Table 3, widths 64 and 128, it seems that MLP has better accuracy than some KAN based methods.

Response 2: Thank you very much for this observation. For the Houston 18 image, we significantly (4 times) increased the maximum number of epochs during training for both KAN and MLP. As a result, we obtained values for KAN that are indistinguishable from or superior to MLP. To ensure that all results in the above table correspond to uniform conditions, we recalculated the above table completely with an expanded number of epochs. Thus, we updated the experimental data in the text of the article.

Comments 3: Spellings could be checked once again, for example line 77, convolved should be convoluted.

Response 3: Thank you for your comment. We replaced “convolved” with “convoluted”.

Comments 4: Overall, this is a very nicely written manuscript and could be accepted without any modifications.

Response 4: Thank you again for your comments and appreciation of our manuscript. We hope that the revised version of the article meets the high standards of the journal.

Round 2

Reviewer 1 Report

Comments and Suggestions for Authors

The form of figures can be midified for better presentation quality. However, the current version is enough for publication.